# PRUNING WITH HINTS: AN EFFICIENT FRAMEWORK FOR MODEL ACCELERATION

## ABSTRACT

In this paper, we propose an efficient framework to accelerate convolutional neural networks. We utilize two types of acceleration methods: pruning and hints. Pruning can reduce model size by removing channels of layers. Hints can improve the performance of student model by transferring knowledge from teacher model. We demonstrate that pruning and hints are complementary to each other. On one hand, hints can benefit pruning by maintaining similar feature representations. On the other hand, the model pruned from teacher networks is a good initialization for student model, which increases the transferability between two networks. Our approach performs pruning stage and hints stage iteratively to further improve the performance. Furthermore, we propose an algorithm to reconstruct the parameters of hints layer and make the pruned model more suitable for hints. Experiments were conducted on various tasks including classification and pose estimation. Results on CIFAR-10, ImageNet and COCO demonstrate the generalization and superiority of our framework.

## 1 INTRODUCTION

In recent years, convolutional neural networks (CNN) have been applied in many computer vision tasks, *e.g.* classification Krizhevsky & Hinton (2009); Deng et al. (2016), objects detection Everingham et al. (2010); Ren et al. (2015), and pose estimation Lin et al. (2014). The success of CNN drives the development of computer vision. However, restricted by large model size as well as computation complexity, many CNN models are difficult to be put into practical use directly. To solve the problem, more and more researches have focused on accelerating models without degradation of performance.

Pruning and knowledge distillation are two of mainstream methods in model acceleration. The goal of pruning is to remove less important parameters while maintaining similar performance of the original model. Despite pruning methods' superiority, we notice that for many pruning methods with the increase of pruned channel number, the performance of pruned model drops rapidlly. Knowledge distillation describes teacher-student framework: use high-level representations from teacher model to supervise student model. Hints method Romero et al. (2015) shares a similar idea of knowledge distillation, where the feature map of teacher model is used as high-level representations. According to Yosinski et al. (2014), the student network can achieve better performance in knowledge transfer if its initialization can produce similar features as the teacher model. Inspired by this work, we propose that pruned model outputs similar features with original model's and provide a good initialization for student model, which does help distillation. And on the other hand, hints can help reconstruct parameters and alleviate degradation of performance caused by pruning operation. Figure 1 illustrates the motivation of our framework. Based on this analysis, we propose an algorithm: we do pruning and hints operation iteratively. And for each iteration, we conduct a reconstructing step between pruning and hints operations. And we demonstrate that this reconstructing operation can provide a better initialization for student model and promote hints step (See Figure 2). We name our method as PWH Framework. To our best knowledge, we are the first to combine pruning and hints together as a framework.

Our framework can be applied on different vision tasks. Experiments on CIFAR-10 Krizhevsky & Hinton (2009) , ImageNet Deng et al. (2016) and COCO Lin et al. (2014) datasets show the

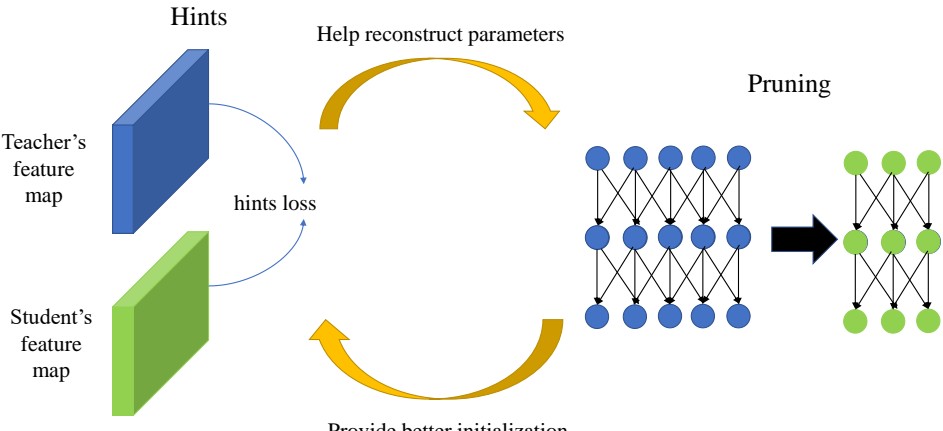

Figure 1: Motivation of PWH Framework. The pruning and hints are complementary to each other. Hints can help pruned model reconstruct parameters. And the network pruned from the teacher model can provide a good initialization for student model in hints learning.

effectiveness of our framework. Furthermore, our method is a framework where different pruning and hints methods can be included.

To summarize, the contributions of this paper are as follows: (1) We analyze the properties of pruning and hints methods and show that these two model acceleration methods are complementary to each other. (2) To our best knowledge, this is the first work that combines pruning and hints. Our framework is easy to be extended to different pruning and hints methods. (3) Sufficient experiments show the effectiveness of our framework on different datasets for different tasks.

## 2 RELATED WORK

Recently, model acceleration has received a great deal of attention. Quantization methods Rastegari et al. (2016); Courbariaux et al. (2016; 2015); Juefei-Xu et al. (2017); Zhou et al. (2017)reduce model size by quantizing float parameters to fixed-point parameters. And fixed-point networks can be speeded up on special implementation. Group convolution based methods Howard et al. (2017); Chollet (2017) separates a convolution operation into several groups, which can reduce computation complexity. Several works exploit linear structure of parameters and approach parameters using low-rank way to reduce computational parameters Denton et al. (2014); Jaderberg et al. (2014); Alvarez & Petersson (2016); Zhang et al. (2016). In our experiments, we use two of current mainstream model acceleration way: pruning and knowledge distillation.

### 2.1 PRUNING

Network pruning has been proposed for a long time, such as Hanson & Pratt (1989); Hassibi et al. (1993); LeCun et al. (1990). Recent pruning methods can be roughly adopted in two levels, *i.e.* channel-wise Molchanov et al. (2017); He et al. (2017); Sajid Anwar (2015); Hu et al. (2017) and parameter-wise Han et al. (2016; 2015); Yang et al. (2017); Li et al. (2017a); Wen et al. (2015); Luo et al. (2017).In this paper, we use channel-wise approach as our pruning method. There are many methods in channel-wise family. He *et al.* He et al. (2017) prune channels in LASSO regression way from sample feature map. Proposed in Liu et al. (2017), the scale parameters in Batch Normalization layers are used to evaluate the importance of different channels. Molchanov *et al.* Molchanov et al. (2017) use taylor formula to analyze each channel's contribution to the loss and prune the lowest contribution channel. We utilize this method in our framework. Despite the superiority of pruning

methods, we find that the effectiveness of them will observably decrease with the increase of pruned channel numbers.

## 2.2 DISTILLATION

Knowledge distillation (KD) Hinton et al. (2015) is the pioneering work of this field. Hinton *et al.* Hinton et al. (2015) define soft targets and use it to supervise student networks. Beyond soft targets, hints are introduced in Fitnets Romero et al. (2015), which can be explained as whole feature map mimic learning. Several researches have focused on hints. Zagoruyko *et al.* Zagoruyko & Komodakis (2017) propose atloss to mimic combined output of an ensemble of large networks using student networks. Furthermore, Li *et al.* Li et al. (2017b) demonstrate a mimic learning strategy based on region of interests to improve small networks' performance for object detection. However, most of these works train student model from scratch and ignore the significance of student networks' initialization.

## 3 OUR APPROACH

In this section, we will describe our method in details. First we show hints and pruning methods which have been used in our framework. Then we introduce reconstructing operation and analyze its effectiveness. Finally, combining hints, pruning and reconstructing operation, we propose PWH Framework.

## 3.1 PRUNING STEP

The pruning method we use in this paper is based on Molchanov et al. (2017). The algorithm can be described as a combinatorial optimization problem:

$$\min_{W'} \left| C(D|W') - C(D|W) \right| \quad st \quad ||W'||_0 \leq B, \tag{1}$$

Where $C(\cdot)$ is the cost function of the task, $D$ is the training samples, $W$ and $W'$ are the parameters of original and pruning networks. In the optimization problem, $||W'||_0$ bounds the number of non-zero parameters in $W'$. The parameter $W_i$ whether to be pruned completely depends on its outputs $h_i$. And the problem can be redescribed as minimizing $\left| \Delta C(h_i) \right| = \left| C(D, h_i = 0) - C(D, h_i) \right|$. However, it's hard to find global optimal solution of the problem. Using taylor expansion can get approximate formula of the objective function:

$$
\begin{aligned}
C(D, h_i = 0) &\approx C(D, h_i) - \frac{\delta C}{\delta h_i} h_i \\
\left| \Delta C(h_i) \right| &\approx \left| C(D, h_i) - \frac{\delta C}{\delta h_i} h_i - C(D, h_i) \right| = \left| \frac{\delta C}{\delta h_i} h_i \right|.
\end{aligned}
\tag{2}
$$

During backpropagation, we can get gradient $\frac{\delta C}{\delta h_i}$ and activation $h_i$. So this criteria can be easily implemented for channel pruning.

## 3.2 HINTS STEP

Hints can provide an extra supervision for student network, and it usually combines with task loss. The whole loss function of hints learning is represented as follows:

$$L = L_t + \lambda L_h \tag{3}$$

Where $L_t$ is the task loss (*e.g.* classification loss) and $L_h$ is the hints loss. $\lambda$ is hints loss weight which determines the intensity of hints supervision. There are many types of hints methods. Different hints methods are suitable for different tasks and network architectures. To demonstrate the superiority and generalization of our framework, we try three kinds of hints methods in our experiments: L2 hints, normalization hints and adaption hints. We introduce L2 hints, normalization hints in appendix.

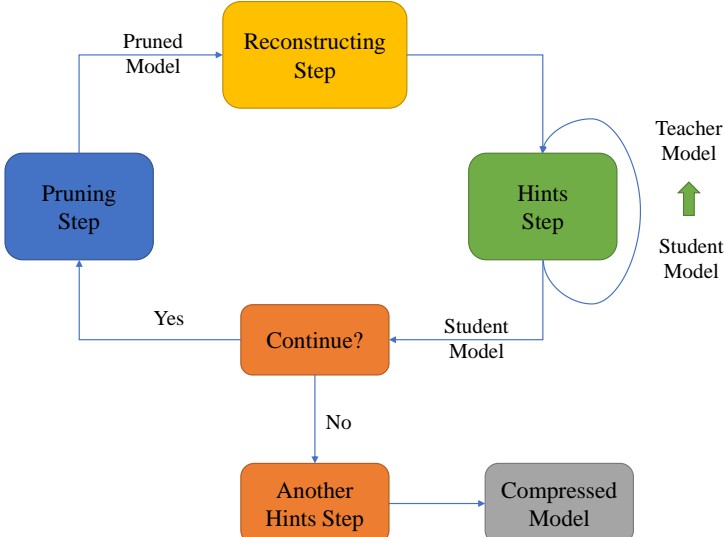

Figure 2: The pipeline of PWH Framework. The whole framework is composed of three steps. First we slim the original network with reducing certain number of channels. Then we reconstruct the hints layer of the pruned model to minimize the difference of feature map between teacher and student. Finally, we start hints step to advance the performance of pruned model.

**Adaption Hints**   Chen et al. (2017) demonstrates that it's necessary to add an adaption layer between student and teacher networks. The adaption layer can help transfer student layer feature space to teacher layer feature space, which will promote hints learning. The expression of adaption hints is described in equation 4.

$$L_m = \frac{1}{N} \|f_t - r(f_s)\|_2^2 \tag{4}$$

Where $r(\cdot)$ is adaption layer. And for convolutional neural networks, adaption layer is $1 \times 1$ convolution layer.

### 3.3   RECONSTRUCTING STEP

The objective function of reconstructing step is:

$$\min_W \|Y - WX\|_2^2 \tag{5}$$

Where $Y$ is the feature map of original (teacher) model, $X$ is the input of hints layer and $W$ is the parameter of hints layer. The optimization problem has close-form solution using least square method: $W = (X^{\mathrm{T}}X)^{-1}X^{\mathrm{T}}Y$. However, because some datasets (*e.g.* ImageNet Deng et al. (2016)) have numerous images, $X$ will be high-dimension matrix. And it's impossible to solve the problem which involves such huge matrix in one time. So we randomly sample images in dataset and compute $X$ according to these images. Due to the randomness, the reconstructed weights may be worse than original weights. Thus, we finally use a switch to select better weights (See equation 6).

$$W_f = \arg\min_W \|Y_0 - WX_0\|_2^2 \quad W \in \{W_o, W_r\} \tag{6}$$

Where $W_f$, $W_o$ and $W_r$ are the final weights, original weights and reconstructed weights of hints layer. $Y_0$ and $X_0$ are computed from the whole dataset. The objective function of Normalized L2 loss (See equation 16) is different from L2 loss, but we explain that the reconstruction step is also effective for normalized L2 loss. The details of proof is available in supplementary material.

### 3.4   PWH FRAMEWORK

Combining pruning step, reconstructing step and hints step, we propose our PWH Framework. The framework iteratively conducts three steps. For pruning, we reduce the model size by certain num-

bers of channels. Then the parameters of hints layer will be reconstructed to minimize the difference of feature map between pruned model's and teacher model's. Finally we use pruned model as student, original model as teacher and conduct hints learning. And in the next iteration, the original model will be substituted by the student model in this iteration. After training, the student model becomes the teacher model for the next iteration. And another hints step is implemented at the ending of the framework where the teacher model will be set as the original model at the beginning of training (the teacher model in the first iteration). The pseudocode of our approach is provided in appendix.

We demonstrate that compared with the original model in the first iteration, the student model in the current iteration is a better candidate for the teacher model in next hints step. The reason is that the model before pruned and after pruned have more similar feature map and parameters, which can promote and speed up hints step. At the end of the whole framework, we do another hints step. Different from preceding hints step, the teacher model is selected as the original model in the first iteration. We demonstrate that the final hints step is like the finetune step in pruning methods, which may need long-time training and improves the performance of compressed model. And original model in the first iteration will be the better teacher. Figure 2 shows the pipeline of our framework.

## 3.5 ANALYSIS

The hints and pruning in PWH Framework are complementary to each other. On one hand, pruned model is a good initialization to student model in hints step. We propose that the feature map of pruned model is similar to original model's compared with random initialization model's. In this way, proposed in Yosinski et al. (2014), the transferability between student and teacher network will increase, which is beneficial for hints learning. Experiments in §4.4 demonstrate that the difference between original model's and pruned model's feature map is much smaller than random initialized model's. On the other hand, hints helps pruning reconstruct parameters. We demonstrate that when pruned channels number is large, pruning method is inefficient. The pruning operation will bring large degradation of performance in this case. We find that pruning numerous channels will destroy the main structure of networks(See 3). And hints can alleviate this trend and help recover the structure and reconstruct parameters in model(See 4).

The motivation of reconstruct step is the generalization of our method. Our approach is a framework and it should be available for different pruning methods. However, the criteria of some pruning methods are not based on minimizing the reconstructing error of feature map . In other words, there is still room to improve the similarity between feature map of original (teacher) and pruned (student) networks, which is beneficial for hints learning. We only conduct reconstructing operation on hints layer because it can not only reduce the difference of feature map used for hints but also maintain the main structure of the pruned model (See experiments in 4.3.3). Moreover, for adaption hints methods, it need to initialize adaption layer(hint layer). Compared with random initialization, reconstruction operation can help to construct this layer and provide more similar features with teacher models'.

## 4 EXPERIMENTS

We conduct experiments on CIFAR-10 Krizhevsky & Hinton (2009) , ImageNet Deng et al. (2016) and COCO Lin et al. (2014) for classfication and pose estimation tasks to demonstrate the superiority of PWH Framework. In this section, we first introduce implementation details in different experiments on different datasets. Then we compare our method with pruning methods and hint methods. Furthermore, in §4.3 we do ablation study to further analyze the framework. Finally we analyze the effectiveness of our approach in §4.4 and show hints and pruning methods are complementary to each other.

## 4.1 IMPLEMENTATION DETAILS

We train networks using PyTorch deep learning framework. Pruning-only refers to a classical iterative pruning and finetuning operation. And for hints-only methods, we set original model as teacher model and use the compressed random initialized model as student model. For fair comparison, the student model in hints-only shares the same network structure with student model in PHW Frame-

Table 1: The main results for PWH Framework. 'baseline' means the performance of the original model. '50% pruned' in column-4 denotes that compressed model's channel number is 50% less than original's. Results on different network architectures for different tasks demonstrate the superiority of PWH Framework. Our approach outperforms hints-only and pruning-only for a large margin.

| Dataset | Model | Baseline | Pruned | Pruning-only | Hints-only | PWH Framework |
|---|---|---|---|---|---|---|
| ImageNet | ResNet18 | 69.76 | 30% | 66.01 | 64.98 | **67.35** |
| CIFAR-10 | VGG16BN | 92.72 | 60% | 91.67 | 90.71 | **92.5** |
| COCO | ResNet18 FPN | 68.4 | 30% | 66.3 | 60.9 | **67.4** |

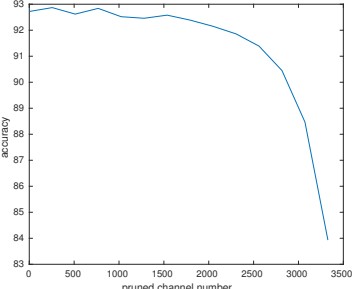

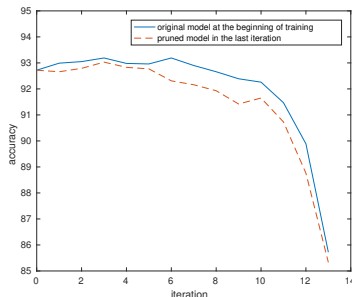

Figure 3: The relationship between the performance of pruned model and the number of pruned channels

Figure 4: The comparison for different selections of teacher model in PWH Framework. The teacher model in this experiment is set as the original model at the beginning of training or the pruned model in the previous iteration.

work. We use the standard SGD with momentum set to 0.9. The standard weight decay is set to 1e-4. VGG-16 Simonyan & Zisserman (2015) network with BatchNorm Ioffe & Szegedy (2015), ResNet18 He et al. (2016) and ResNet18 with FPN Chen et al. (2018) are used for CIFAR-10, ImageNet and COCO respectively.

## 4.2 MAIN RESULTS

Table 1 illustrates results. We can find that PWH Framework outperforms pruning-only method and hints-only method for a large margin on all datasets for different tasks, which verify the effectiveness of our framework and also shows that hints and pruning can be combined to improve the performance. Results on different tasks and models show that PWH Framework can be implemented without the restriction of tasks and network architectures. Moreover, illustrated in table 1, our framework can be applied for different pruning ratios, which means that we can adjust pruning ratio in the framework for different tasks to achieve different acceleration ratios.

## 4.3 ABLATION STUDY

To further analyze PWH Framework, we do several ablation studies. All experimetns are conducted on CIFAR-10 dataset using VGG16BN. The feature map proposed in this section refers to the output of last convolution layer, which is also the feature map used for hints learning. And in this section, we do ablation study for three different aspects. First, we do experiments to show iterative operation is an important component of PWH Framework. Then we study on the selection of teacher model in hints step. Finally, we validate on the effects of reconstructing step.

### 4.3.1 THE IMPORTANCE OF ITERATION

In PWH Framework, we implement three steps iteratively. And in this section we will show the importance and necessity of iterative operation. We conduct an experiment to compare the effects

Table 2: The comparison of the reconstructing feature map error between pruned model and random initial model.

| Pruned Number | Pruned Model | Random Model |
|---|---|---|
| 256 | 1.84 | 60.93 |
| 512 | 4.97 | 65.15 |
| 768 | 9.02 | 64.45 |
| 1024 | 13.58 | 62.2 |

Table 3: The effects of iterative operation. '70% pruned' in column-4 denotes that compressed model's channel number is 70% less than original's.

| Dataset | Model | Pruned | Method | Accuracy |
|---|---|---|---|---|
| CIFAR-10 | VGG16BN | 70% | once | 89.4 |
| | | | iterative | **90.16** |

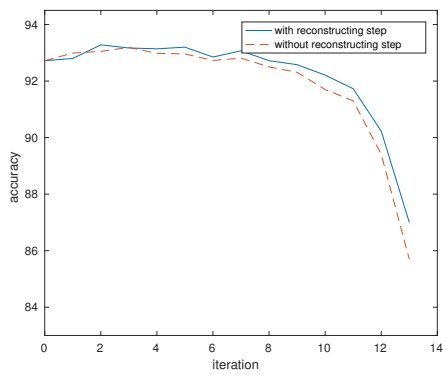

Figure 5: The experiment on verifying the effectiveness of reconstructing step. The Figure shows comparison of using and without using reconstructing step for accuracy.

Figure 6: The relationship between the performance of network and the number of pruned channels using different methods. We conduct experiment iteratively and for each iteration we prune 256 channels.

of doing pruning and hints only once (*i.e.* First do pruning and then do hints. Both operations are conducted only once.) and doing pruning and hints iteratively. Table 2 shows results.

We can see that iterative operation can improve the performance of model dramatically. To further explain this result, we do another experiment: we analyze the relationship between the performance of pruned model and the number of pruned channels. Results in Figure 3 illustrate that when the number of pruned channels is large, the performance of pruned model will drop rapidlly. Thus, if we only do pruning and hints once, pruning will bring large degradation of performance and pruned model cannot output the similar feature to original model's. And in this way, pruned model is not a more resonable initialization to student model. Pruning step is useless to hints step in this situation.

### 4.3.2 THE SELECTION OF TEACHER MODEL

The teacher model is the pruned model from previous iteration in PWH Framework. Original model at the beginning of training can be another choice for teacher model in each iteration. We do an experiment to compare these two set-up for teacher model. And in this experiment, we prune 256 channels in each iteration. Figure 4 shows results.

We observe that when iteration is small, using original model in the first iteration as teacher model has a comparable performance with using the pruned model in the previous iteration. However, with the increase of iterations, we can find that superiority of using the pruned model in the previous iteration increases. The reason is that the original model in the first iteration has higher accuracy so it performs well when iteration is small. But when iteration becomes large, pruned model's feature map will have large difference with original model's feature map. And in this situation, there is a gap between pruned model and teacher model in hints step. On the other hand, using the pruned model in the previous iteration will increase the similarity of feature map between student model's and teacher model's, which will help distillation in hints step.

### 4.3.3 THE EFFECTIVENESS OF RECONSTRUCTING STEP

Proposed in §3.3, reconstructing step is used to further refine pruned model's feature and make it more similar to teacher's. We conduct the experiment to validate the effectiveness of reconstructing step. To fairly compare, we implement PWH Framework with and without reconstructing step. In each iteration, we prune 256 channels. We study on the accuracy of compressed model using two different methods. Furthermore, we also analyze L2 loss between pruned model's and original model's feature map in each iteration. Figure 5 shows experiment results. We find that the method with reconstructing step performs better. We want our framework to be adaptive to different pruning methods but some of the pruning methods' criteria are not minimizing the reconstructing feature map's error. Reconstructing step can be used to solve this problem and increase the similarity of feature maps between two models.

## 4.4 ANALYSIS OF PWH FRAMEWORK

To further analyze the properties of PWH Framework, we conduct further experiments on our approach. The experiments results verify our assumptions: pruning and hints are complementary to each other. All experiments are conducted on CIFAR-10 dataset using VGG16 as the original model.

### 4.4.1 PRUNING HELPS HINTS

We conduct experiment to compare the reconstructing feature map error between pruned model and random initial model. We use the pruning method in §3.1 to prune certain number of channels from original network and we calculate L2 loss between pruned model's feature map and original model's feature map. Similarly, we randomly initialize a model whose size is same to the pruned model and calculate L2 loss of feature map between this model and original model. Then we increase pruned channel number and record these two errors accordingly. In Table 2, we notice that in a large range (0-1024 pruned channels) pruned model's feature map is much more similar to original model's. And this demonstrate that the transferability between pruned model and original model is larger. And student model, initialized with the weights of pruned model, can perform better in hints learning.

### 4.4.2 HINTS HELP PRUNING

To demonstrate hints method is beneficial to pruning, we first compare experiments between pruning-only with *pruning and hints*. Different from PWH Framework, pruning and hints method used in this section doesn't have reconstructing step. This is because we want to show the effectiveness of hints and reconstructing step is an extraneous variable. In contrast experiments, we iteratively implement pruning and hints operations . To fairly compare, in pruning and hints method, we substitute finetune operation for hints operation to get pruning-only method. In Figure 6, we observe that pruning and hints method has comparable performance with pruning-only method on small amount of iteration. However, the margin of two methods becomes larger and larger with the increase of iterations (pruned channels number). This phenomenon caused by the huge performance degradation in pruning operation when the original model is small. The small model doesn't have many redundant neurons and the main structure will be broken in pruning. And hints can alleviate this trend and help reconstruct parameters in pruned model.

## 5 CONCLUSION

In this paper, we propose PWH Framework, an iterative framework for model acceleration. Our framework takes the advantage of both pruning and hints methods. To our best knowledge, this is the first work that combine these two model acceleration methods. Furthermore, we conduct reconstructing operation between hints and pruning steps as a cascader. We analyze the property of these two methods and show they are complementary to each other: pruning provides a better initialization for student model and hints method helps to adjust parameters in pruned model. Experiments on CIFAR-10, ImageNet and COCO datasets for classification and pose estimation tasks demonstrate the superiority of PWH Framework.

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

APPENDIX

In this supplementary material, we first provide more implementation details for better illustration of our experiments. In the second part, we give a proof in §3.3. We show that the upper bound of normalized L2 loss will decrease if L2 loss decreases theoretically.

## A    IMPLEMENTION DETAILS

Following section contains more implementation details and We use PyTorch deep learning framework with 4 NVIDIA Titan X GPUs.

### A.0.1    **CIFAR-10:**

CIFAR-10 dataset has 10 classes containing 6000 $32 \times 32$ color images each. 50000 images are used for training and 10000 for test. We use top-1 error for evaluation. For model, we use the VGG-16 Simonyan & Zisserman (2015) network with BatchNorm Ioffe & Szegedy (2015). In CIFAR-10 finetune (hints) step, we use standard data augmentation with random cropping with 4 pixels, mean substraction of (0.4914, 0.4822, 0.4465) and std to (0.2023, 0.1994, 0.2010). A batch size of 128 and learning rate of 1e-3 are used . We set finetune (hints) epoch as 20. The hints method utilized on this dataset is L2 hints method with loss weight 10. We sample 1000 images to reconstruct weights.

### A.0.2    **IMAGENET:**

ImageNet classification dataset consists of 1000 classes. We train models on 1.28 million training images and test models on 100k images. Top-1 error is used to evaluate models. In this experiment, ResNet18 He et al. (2016) is our original model and we use PWH Framework to compress it. During finetune (hints) stage, the batchsize is 256, learning rate is 1e-3. We set mean substraction to (0.485, 0.456, 0.406) and std to (0.229, 0.224, 0.225). The loss weight is set as 0.5. The random cropping is used. In rescontructing stage, 1000 images are sampled to reconstruct hints layer weights.

### A.0.3    **COCO:**

We conduct pose estimation experiment on COCO dataset. In this experiment, we train our models on trainval dataset and evaluate models on minival set. The evaluation criteria for COCO dataset we use is OKS-based mAP. We use ResNet18 with FPN Chen et al. (2018) as the original model in the experiment. And in this experiment, we use random cropping, random rotation and random scale as our data augmentation strategy. We use a weight decay of 1e-5 and learning rate of 5e-5. The loss weight is set as 0.5. A batch size of 96 is used. The number of sampled images in reconstructing step is 500.

## B    THE PROOF IN §3.3

In reconstructing step, we use least square method to reconstruct parameters in hints layer. The objective function of this step can be described in equation 7.

$$\min_{W} \|Y - WX\|_2^2 \tag{7}$$

Where $Y$ is the feature map of original (teacher) model, $X$ is the input of hints layer and $W$ is the parameter of hints layer. However, many hints methods use normalized L2 loss as hints loss (See equation 8). It's difficult to optimize the problem using common methods if we set normalized L2 loss as objective function.

$$\min_{W} \left\| \frac{Y}{\|Y\|} - \frac{WX}{\|WX\|} \right\|_2^2 \tag{8}$$

In this section, we will show that the upper bound of normalized L2 loss is related to L2 loss. In other words, if L2 loss decreases, the upper bound of L2 loss will decrease. For an image $x$, $y$ is the feature map of teacher network whose input is $x$. We suppose that:

$$y = Wx + e = y_0 + e$$
$$E[\|y - y_0\|^2] = E[\|e\|^2] \leq M \tag{9}$$

Where $e$ is the error and it is independent with $x$. $E[\cdot]$ means the expectation. We suppose that $\frac{1}{\|y\|}$ can be expressed as the function of $y_0$ and $e$ using taylor expansion. Equation 10 shows the expression.

$$\frac{1}{\|y\|} = \frac{1}{\|y_0 + e\|} = \frac{1}{\|y_0\|} + A^{\mathrm{T}}e + o(A^{\mathrm{T}}e) \approx \frac{1}{\|y_0\|} + A^{\mathrm{T}}e \tag{10}$$

Where $A^{\mathrm{T}}$ is the Jacobian matrix. And we use this approximation to estimate the upper bound of normalized L2 loss:

$$
\begin{aligned}
&E[\left\|\frac{y}{\|y\|} - \frac{y_0}{\|y_0\|}\right\|^2] \\
=&E[\left\|\frac{y_0 + e}{\|y_0 + e\|} - \frac{y_0}{\|y_0\|}\right\|^2] \\
\approx&E[\left\|\frac{e}{\|y_0\|} + (y_0 + e)(A^{\mathrm{T}}e)\right\|^2] \\
=&E[\left\|(\frac{1}{\|y_0\|}I + y_0 A^{\mathrm{T}} + e A^{\mathrm{T}})e\right\|^2]
\end{aligned}
\tag{11}
$$

Where $I$ is the identity matrix. For convenience, we assume that $K = \frac{1}{\|y_0\|}I + y_0 A^{\mathrm{T}}$. And $K$ is irrelevant to $e$. The above equation can redescribed as:

$$
\begin{aligned}
=&E[\|(K + e A^{\mathrm{T}})e\|^2] \\
=&E[e^{\mathrm{T}}(K + e A^{\mathrm{T}})^{\mathrm{T}}(K + e A^{\mathrm{T}})e] \\
=&E[e^{\mathrm{T}}K^{\mathrm{T}}Ke + 2e^{\mathrm{T}}Ae^{\mathrm{T}}Ke + e^{\mathrm{T}}Ae^{\mathrm{T}}e A^{\mathrm{T}}e]
\end{aligned}
\tag{12}
$$

Using Cauchy Inequality and the property of eigenvalue, we propose that:

$$
\begin{aligned}
e^{\mathrm{T}}K^{\mathrm{T}}Ke &\leq \lambda_1 e^{\mathrm{T}}e = \lambda_1 \|e\|^2 \\
(e^{\mathrm{T}}A)(e^{\mathrm{T}}Ke) &\leq (\|e\|\|A\|)(\lambda_2 e^{\mathrm{T}}e) = \lambda_2 \|A\| \|e\|^3 \\
e^{\mathrm{T}}Ae^{\mathrm{T}}e A^{\mathrm{T}}e &\leq |e^{\mathrm{T}}A||e^{\mathrm{T}}e||A^{\mathrm{T}}e| \leq \|A\|^2 \|e\|^4
\end{aligned}
\tag{13}
$$

Where $\lambda_1$ and $\lambda_2$ are the maximal eigenvalues of $K^{\mathrm{T}}K$ and $K$. We suppose that $\|e\| \leq 1$. In this way, we can get the estimation of normalized L2 loss:

$$
\begin{aligned}
&E[\left\|\frac{y}{\|y\|} - \frac{y_0}{\|y_0\|}\right\|^2] \\
\leq&\lambda_1 E[\|e\|^2] + \lambda_2 E[\|A\| \|e\|^3] + E[\|A\|^2 \|e\|^4] \\
=&\lambda_1 E[\|e\|^2] + \lambda_2 E[\|A\|]E[\|e\|^3] + E[\|A\|^2]E[\|e\|^4] \\
\leq&(\lambda_1 + \lambda_2 E[\|A\|] + E[\|A\|^2])M
\end{aligned}
\tag{14}
$$

And $A$ is irrelevant to $M$. So we conclude that when L2 loss $M$ decreases, the upper bound of normalized L2 loss will decrease.

## C    HINTS METHODS USED IN EXPERIMENTS

**L2 Hints**    L2 hints is a widely used hints method. It can be easily implemented by adding an extra L2 loss. L2 hints use the euclidean distance of feature map between teahcer and student model's as the supervision. Equation 15 shows the expression of L2 hints.

$$L_m = \frac{1}{N} \|f_t - f_s\|_2^2 \tag{15}$$

Where $f_t$ and $f_s$ are the feature map of teacher and student networks. $N$ is the total number of elements in feature map.

**Normalization Hints**   Unlike normal hints loss, normalization hints first do normalization operation to both teacher's and student's features and then calculate the euclidean distance of two normalized features. Equation 16 shows the expression of normalization hints.

$$L_m = \frac{1}{N} \left\| \frac{f_t}{\|f_t\|} - \frac{f_s}{\|f_s\|} \right\|_2^2 \tag{16}$$

## D   PSEUDOCODE

The pseudocode of our method is as follows:

---
**Algorithm 1** The pipeline of PWH Framework

---
**Input:**  Initial original model $W_0$, Dataset $X$
**Output:**  Compressed model $W$
  1: Iterations $T$, channels to be pruned in an iteration $C$, student model $W_s$, teacher model $W_t$
  2: $W_p = P(W, C, X)$, $P(\cdot)$ represents pruning step
  3: $W_h = H(W_t, W_s, X)$, $H(\cdot)$ represents hints step
  4: $W_r = R(W, X)$, $R(\cdot)$ represents reconstructing step
  5: Initial $W_t = W_0$, $i = 1$
  6: **while** $i <= T$ **do**
  7:     $W_p \leftarrow P(W_t, C, X)$
  8:     $W_s \leftarrow R(W_p, X)$
  9:     $W \leftarrow H(W_t, W_s, X)$
10:     $W_t \leftarrow W$
11: **end while**
12: $W \leftarrow H(W_0, W, X)$

---

