# OpenReview forum: "PRUNING WITH HINTS: AN EFFICIENT FRAMEWORK FOR MODEL ACCELERATION"
_ICLR.cc/2019/Conference_

### Official Review · AnonReviewer1 · 2018-11-01
**interesting combination of two methods but low on novelty**

**Rating:** 4
**Confidence:** 4

**Review:**

This paper uses pruning and model distillation iteratively to reduce the model sizes. The pruning step is based on Molchanov et al. (2017). This is followed by a hints step that minimizes the feature map difference between student and teacher. Finally, a reconstruction step is used to restore original weights. Results are shown on CIFAR-10, Imagenet and COCO datasets for classification and pose estimation tasks where PWH reduces model costs with a small loss in accuracy.

The paper is interesting that it proposes a unique combination of existing methods iteratively to improve the compression rates in modern CNNs. However, given that these methods already exist, the novelty aspect of this paper is low. Furthermore, it is also hard to rebut these methods, since they have been published and extensively evaluated in the respective papers. Nevertheless, it is interesting to note that both methods assist one another in your set of experiments.

In order to improve the paper and for the reviewers to judge the papers more favorably, the authors can compute the time for reconstruction/hints and demonstrate that it is clearly superior to fine-tuning/re-training and offers a clear advantage. This should be emphasized and articulated in related work and introduction and will help the paper.

How did you arrive at the magic number of pruning just 256 channels in every step?

---

### Official Review · AnonReviewer3 · 2018-11-06
**a pruning-with-hints framework for model acceleration**

**Rating:** 5
**Confidence:** 4

**Review:**

In this paper, the authors propose a pruning-with-hints framework for model acceleration.  Via performing pruning and hints iteratively, one can leverage the complementary characteristics of these two approaches to boost performance.

Here are the comments:
1 The framework seems to be technically sound. However, the novelty is limited. Most techniques (e.g., pruning, hints) have been widely investigated in the literature. Reconstruction can be treated as another type of hints. Furthermore, the integration of these strategies is standard.
2 In the experiment, the pruning rate is relatively low for larger models and data sets.  Hence, the effectiveness should be further investigated.  Additionally, please compare with the state-of-the-art approaches such as light-weight design (e.g., MobileNet, ShuffleNet).  This can further enhance the motivation of choosing the proposed framework for real-life applications.

---

### Official Review · AnonReviewer4 · 2018-11-13
**Propose a model to combine some existing techniques for model acceleration.**

**Rating:** 4
**Confidence:** 3

**Review:**

This paper proposes a new framework which combines pruning and model distillation techniques for model acceleration. Though the ``pruning” (Molchanov et al. (2017)) and hint components already exists, the authors claim to be the first to combine them, and experimentally show the benefit of jointly and iteratively applying the two techniques. The authors show better performance of their new framework over baseline, pruning only method and hint only method on a few standard Vision data set.

The motivation is clearly conveyed in the paper. As a framework of combining two existing techniques, I expect the framework can stably improve its two components without too much additional time cost. I have some small questions.

--What is the ``additional cost” of your proposed framework. For example, how many iterations do you typically use. For each data set, what time delta you spent to get the performance improvement comparing to pruning only or hint only models.
--In your iterative algorithm (pseudo code in appendix), the teacher model is only used in the very beginning and final step, though richest information is hidden in the original teacher model. In the intermediate steps, you are fine tuning iteratively without accessing the original teacher model.
--In your reconstruction step, you said due to the randomness, you do not always use the learned new W. How much your algorithm benefit from this selection strategy?

---

### Meta-Review · Area_Chair1 · 2018-12-17
**lack novelty**

**Confidence:** 5
**Recommendation:** Reject

**Metareview:**

This paper proposes a new framework which combines pruning and model distillation techniques for model acceleration. The reviewers have a consensus on rejection due to limited novelty.